# Handling discontinuities in numerical ODE methods for Lagrangian oceanography

Jenny M. Mørk<sup>1,2</sup>, Tor Nordam<sup>2,1</sup>, and Siren Rühs<sup>3</sup>

<sup>1</sup>Department of Physics, Norwegian University of Science and Technology, Trondheim, Norway

<sup>2</sup>SINTEF Ocean, Trondheim, Norway

<sup>3</sup>Leibniz Institute for Baltic Sea Research Warnemünde (IOW), Rostock, Germany

Correspondence: Jenny M. Mørk (jenny.m.mork@ntnu.no)

**Abstract.** In Lagrangian oceanography, numerical methods for Ordinary Differential Equations (ODEs) are used to model particle transport. In many common applications, the velocity field driving the particle transport is provided as output from ocean models, on a discrete grid of points. Hence, the velocity field must be interpolated. Depending on the choice of interpolation, the velocity field or its derivatives may have discontinuities. These discontinuities have implications for the accuracy of the numerical ODE methods employed.

We demonstrate that by using information about the location of the discontinuities, we can take these into account, and improve numerical accuracy over standard integration methods that do not take discontinuities into account. The commonly used combination of the fourth-order Runge-Kutta method and linear interpolation of the velocity field, in fact, only yields second-order accuracy with the standard method. By accounting for discontinuities, we can achieve several orders of magnitude better accuracy with the same timestep. The implementation makes use of a combination of known methods from the field of numerical integration of ODEs. The implementation is quite flexible, agnostic to grid layout and order of interpolation, and contributes only modestly to the code complexity. Hence, the proposed technique for handling discontinuities in interpolated velocity fields could easily be adopted to a range of applications where numerical accuracy or efficiency is of importance.

As an example where numerical accuracy is important, we run a backtracking case for particles with known initial conditions, and show that the method with discontinuity handling is to a larger degree able to recover the correct initial positions of the particles, compared to standard fourth-order Runge-Kutta.

#### 1 Introduction

Computations of particle trajectories through pre-calculated velocity fields are frequently encountered, particularly in oceanic and atmospheric transport simulations (van Sebille et al., 2018). Examples include modelling the movement of pollutants such as oil spills (Nordam et al., 2019; Drouin et al., 2019; North et al., 2011), microplastic (Onink et al., 2021, 2019; Kaandorp et al., 2020; Simantiris et al., 2022), and chemicals (Nepstad et al., 2022; Aghito et al., 2023; Povinec et al., 2013), as well as jellyfish (Dawson et al., 2005), algae and plankton (Siegel et al., 2003; Woods, 2005; Visser, 2008), and icebergs (Marsh et al., 2015). Particle trajectory simulations are further used to analyse water volume and air mass transport pathways associated with

the general circulation in the ocean and atmosphere (e.g. Bower et al., 2019; Döös et al., 2017). Similar computations have even been used in studies on the spread of respiratory diseases like COVID-19 (Wilson et al., 2021).

Marine and atmospheric transport applications usually require computation of a large number of particle trajectories. A single trajectory is typically computed by numerically solving an ordinary differential equation (ODE), and computing a large number of trajectories can be quite computationally demanding. Therefore, it is of practical value to have some kind of guidance on how to select numerical schemes to optimise the computations in terms of both accuracy and computational cost.



In recent studies within Lagrangian oceanography one finds a number of different integration methods, ranging from the simple explicit Euler scheme (e.g. Sayol et al., 2014) to higher-order methods, though a common choice seems to be the classic fourth-order Runge-Kutta method with fixed time step (see, e.g. Fedorov et al., 2021; Menezes, 2021; Drouin et al., 2019; Onink et al., 2021). Some also make use of embedded Runge-Kutta pairs with adaptive step size (see, e.g. Guerrini et al., 2021; Simantiris et al., 2022) or the so-called "analytical" methods that ensure volume conservation (for velocity fields from volume-conserving models) or mass conservation (for velocity fields for mass-conserving models) (Blanke and Raynaud, 1997; Döös et al., 2017). The latter methods avoid explicit time-stepping by analytically integrating particle trajectories between grid points, assuming linear interpolation of the velocity fields. A notable common feature is that there is limited discussion around the choice of integration method.

We further note that in marine transport applications, the velocity field in the ODE is represented by ocean currents (and in some cases other fields such as wind or Stokes drift). Advection in the ocean is known to be chaotic, at least on some scales (Koshel and Prants, 2006; Abraham and Bowen, 2002). This means that nearby trajectories tend to separate exponentially over time, and thus numerical errors in trajectory integration will also grow exponentially in time. For that reason it may be useful to resort to higher-order integration methods, or alternatively, low-order methods with small time steps, especially when considering long integration times (months to years or even decades), to limit the accumulation and growth of numerical errors as far as possible.

In many applications, random increments are added to the position or the velocity of the particles, to capture the effects of sub-grid scale diffusion. Formally, one is then solving a Stochastic Differential Equation (SDE), which requires different numerical methods (Kloeden and Platen, 1999). We do not deal with diffusion in the current paper, but focus on representing the advective processes resolved by a given underlying velocity field.

Since the velocity field representing the ocean currents is not continuous, but given at discrete times and spatial locations, it is also necessary to use interpolation to obtain a velocity field that can be used in the integration. Interpolation introduces discontinuities in the velocity field itself or its derivatives, and hence the choice of interpolation scheme may also affect both the accuracy and the computational effort of a given numerical integrator. Nevertheless, there is hardly any discussion around the choice of interpolation scheme in recent publications, and many authors do not even mention it. Among those that do it seems that linear interpolation is the most common choice (e.g. Cividanes et al., 2024; Cunningham et al., 2022; Fifani et al., 2021; Onink et al., 2019), followed by cubic (e.g. Prants et al., 2023; Fedorov et al., 2021; Drouin et al., 2019). In any case, it is not common to state the reasoning behind a choice of interpolation scheme or the implications of that choice.

In this paper, we build on an earlier study of numerical integrators for Lagrangian oceanography (Nordam and Duran, 2020), in which a scheme was introduced to deal with discontinuous partial derivatives along the time dimension. In the current study, we tackle the issue of discontinuous partial derivatives along the spatial dimensions, i.e., when crossing cell boundaries in the velocity field. We investigate the performance of some standard fixed-step Runge-Kutta methods, and compare their performance to modified methods designed to handle the discontinuities in the interpolated velocity field. The methods are compared in terms of performance when combined with different orders of spline interpolation, and we also compare to the earlier results from Nordam and Duran (2020).

In the next section, we will present some theory on the numerical integration of ODEs with interpolated velocity fields. We will introduce the integration and interpolation schemes we use in this study, and give a brief description of how to evaluate the error of a numerical integrator. In Section 3, we discuss the consequence of discontinuities in the interpolated velocity field or its derivatives for the accuracy of numerical integration methods, and present how we modified the integration scheme to handle these discontinuities. Section 4 describes how the numerical experiments were performed, including how the methods were implemented in code. The results of experiments to investigate accuracy are presented in Section 5, along with an example application (backtracking), and a discussion of comparison to other methods. Finally, in Section 6, we present some conclusions.

# 2 Theory


Finding a particle trajectory from a velocity field essentially means solving an ODE on the form

$$\frac{d\mathbf{x}}{dt} = \mathbf{f}(\mathbf{x}, t),\tag{1}$$

where f(x,t) is the particle velocity at position x and time t. When the initial position  $x(t_0) = x_0$  is known one can use numerical integration to find a solution for x(t) for  $t > t_0$ . The concept of numerical integration has been around for a very long time (see Euler (1768, p. 200) for the original description of the forward Euler scheme), and there is a large body of literature on the topic (see, e.g., the classic reference Hairer et al. (2008)). Many different techniques have also been developed, ranging from general methods that are applicable to many different problems (see, e.g., Hairer et al. (2008); Hairer and Wanner (1996)) to more specialised schemes that have been developed to, e.g., respect specific conservation laws (Hairer et al., 2006). In this paper, we will consider selected fixed-step methods from the Runge-Kutta family, including the forward Euler method and the classic fourth-order Runge-Kutta, both of which are commonly used in Lagrangian oceanography. To aid later discussion, we will briefly introduce some notation and elementary concepts from the theory of numerical integration of ODEs. For additional details, the interested reader is referred to the theory presented in Nordam and Duran (2020), which this paper builds on, as well as the general literature (Hairer et al., 2008).

A numerical integration method will solve Eq. (1) by taking repeated discrete steps in time, t. For fixed-step methods t is incremented by the same amount h in every step, so that after n steps we have

$$t_n = t_0 + nh. (2)$$

In this paper, we will use the notation convention where  $x_n$  denotes the numerical approximation to the solution for the time  $t_n$ , while  $x(t_n)$  denotes the true solution for the same time. We assume that there exists such a true solution and that this solution is unique for a given initial condition (Hairer et al., 2008, pp. 35–43), although in most cases where numerical integration is used, the true solution  $x(t_n)$  is unknown.

# 2.1 Error bounds

There are two important measures for the error in numerical integration. One is called the local error and the other is called the global error, and both depend on the timestep, h. The local error is the numerical error in a single step,  $e(h) = x(t_1) - x_1$ , i.e., the difference between the numerical approximation  $x_1$  and the exact solution evaluated at time  $t_1$ , assuming zero error at time  $t_0$ , that is  $x(t_0) = x_0$ . The global error is the total error after N steps, at time  $t_N$ , again assuming zero error at time  $t_0$ . Global error is thus given by (Hairer et al., 2008, p. 159)

$$E(h) = x(t_N) - x_N$$
. (3)

It can be shown (Hairer et al., 2008, p. 157) that, when using a Runge-Kutta method of order p for an ODE  $\dot{x} = f(x,t)$ , where the partial derivatives of f(x,t) up to and including order p exist and are continuous, the local error is bounded by

$$|x(t_1) - x_1| \le Ch^{p+1},$$
 (4)

for some constant C. The constant C will depend on the method, and on f(x,t), but not on the timestep h. If the local error e(h) scales as  $O(h^{p+1})$  then the global error E(h) will scale as  $O(h^p)$  (Hairer et al., 2008, pp. 160–162).

# 2.2 Error estimation



If the exact solution is unknown, the error must be estimated by purely numerical means. Generally, the smaller the step size h of a numerical integration method, the more accurate its solution, and as  $h \to 0$  the numerical solution ideally converges toward the true solution. How fast the numerical solution converges depends on the order, p, of the method, where higher order means faster convergence. The exact solution can thus be approximated by a reference solution computed using a high-order method with a very small timestep  $h_{\rm ref}$  (see Nordam and Duran, 2020).

Note that computers have limited precision when storing numbers and this in turn affects the accuracy of arithmetic with floating-point numbers. For example, in double precision floats, which is probably the most commonly used type for scientific computing, numbers are stored with approximately 16 significant digits, and any operation will tend to pick up a rounding error in the last digit. As a result, when h becomes very small, and the number of operations large, the global error is dominated by accumulated round-off error (see, e.g. Press et al., 2007, p. 10). Reducing the step size further beyond this point will make the error increase rather than decrease, due to the further accumulation of the round-off error when the number of arithmetic operations increases. The size of the time step  $h_{\rm ref}$  must therefore be chosen with some care. We explain how we compute our reference solutions in Appendix C.

# 120 2.3 Interpolation and discontinuities

The velocity fields used in Lagrangian oceanography are typically given by modelled ocean current velocity data at discrete positions and times on a grid. Trajectory computations require a velocity field that can be evaluated at arbitrary positions and times, and such a field can be obtained by interpolating the modelled data. In this study, we have chosen to interpolate the data using spline interpolation, with which the current data is interpolated using several low-order polynomials (de Boor, 2001). We have chosen to consider three spline interpolation schemes of different order:

- second-order: linear interpolation;





- fourth-order: cubic spline interpolation;
- sixth-order: quintic spline interpolation.

Note that there exists more than one definition of the order of interpolation, but the one used here is that the order of interpolation is 1 plus the polynomial degree (de Boor, 2001, p. 1). The key point is that spline interpolation of order m creates an interpolant consisting of a piecewise polynomial function of degree m-1. At the knots, where two different polynomial functions meet, the coefficients of the two polynomials are chosen such that the spline has m-2 continuous derivatives. Away from the knots, all partial derivatives exist and are continuous (with the partial derivatives of order m or higher being zero).

The locations of the discontinuities depend on the interpolation scheme and the grid structure of the data points. For the selected schemes in this study, the discontinuities will be located at the data points<sup>1</sup>. In the temporal dimension (Nordam and Duran, 2020), this is straightforward to handle, but in the spatial dimension, we need to know the grid structure of the velocity data. If the data is given on an Arakawa A-grid, i.e., an unstaggered grid where all velocity vector components are given in the corners of the grid cells, then the discontinuities of the interpolated velocity field will coincide with the hydrodynamic model cell boundaries. If instead the data is on an Arakawa C-grid, i.e., a staggered grid where the velocity components are given on the cell face centers, the discontinuities will have different locations for the different velocity components. That is, if we imagine that the u-points and the v-points are corners of their own respective grids, the interpolated velocity field will have discontinuities along the boundaries of both these grids as illustrated in Fig. 1. Note that this means that velocity fields on C-grids will have twice (in 2D) as many discontinuities as A-grids. When we talk about grid cells in the interpolated velocity field from now on we will be referring to the cells bounded by the discontinuities, not those of the underlying hydrodynamic model. Data that have been produced on a C-grid, but interpolated to cell centers prior to output and storage, are treated the same as A-grid data.

Now, recall from Section 2.1 that the magnitude of the local error of a Runge-Kutta method of order p is only bounded by  $Ch^{p+1}$  when the right-hand side f(x,t) of the ODE has p continuous partial derivatives. This means that we need to be conscious about our choice of interpolation scheme and integrator when setting up a particle transport simulation. Since linear interpolation results in discontinuous first derivatives, in theory not even a first-order integrator is guaranteed to have local error bounded by Eq. (4) when stepping across a grid cell boundary.

<sup>&</sup>lt;sup>1</sup>For other schemes (of odd-numbered order) they can be elsewhere (de Boor, 2001, Ch. VI)

**Figure 1.** Illustration of discontinuities in different grid structures common in hydrodynamic models. If the current data is given on an Arakawa A-grid (left panel) the discontinuities (dashed lines) in the interpolated velocity field coincide with the model cell boundaries (solid lines). If the current data is given on an Arakawa C-grid (right panel) the discontinuities are at the boundaries of the corresponding staggered *u*- and *v*-grids.

# 3 Integrators with discontinuity handling



In this section, we discuss the consequences of the fact that the right-hand side of our ODE has discontinuous derivatives. Furthermore, we present a modification that can be made to any of the commonly used fixed-step Runge-Kutta methods, designed specifically for applications in Lagrangian oceanography (or elsewhere) where interpolated velocity fields are used.

In Section 2.1, we mentioned that in order for it to be guaranteed that a Runge-Kutta method of order p actually is pth-order accurate, the right-hand side of the ODE must have continuous derivatives up to and including order p. If this requirement is not fulfilled, the numerical error may be far larger than expected. As shown in Nordam and Duran (2020), when the error in even just a single step is not bounded by Eq. (4) this can dominate the global error, thus reducing the overall order of the method. In such cases, the use of a higher-order integration scheme is nonsensical as it is just more computationally demanding without the expected benefit of reduced error.

In Section 2.3, we saw that in order to have p continuous derivatives we must have splines of order p+2. A solution could thus be to combine a high-order ODE method with an interpolation scheme of sufficiently high order. However, higher-order spline interpolation is more computationally demanding without necessarily being a more faithful representation of the underlying ocean dynamics. In fact, linear interpolation may be required for particle trajectory simulations based on (i) finite-volume model output or (ii) finite-difference model output, when trajectories are intended to represent volume transport pathways. In these cases, the velocity reconstruction between cell faces must preserve the discrete flux balance, which implies a constant divergence within each grid cell. Linear interpolation satisfies this condition, whereas standard higher-order schemes (e.g.,

cubic or spline interpolation) generally do not. This suggests that a better approach might be to use low-order interpolation together with higher-order ODE methods, but somehow try to handle the discontinuities.

Hairer et al. (2008, pp. 197–198) suggest three alternative ways of dealing with discontinuities in f(x,t) or its derivatives:

- i ignoring them, and letting a variable-step solver adjust the step size to one that gives a small enough error,
- ii using a numerical integration routine that detects and handles discontinuities,




- iii using information about the positions of the discontinuities to stop and restart integration at these locations.
- As mentioned, it is rare to see the implications of the choice of interpolation and integration schemes discussed in applied papers on Lagrangian particle transport. Among those that mention both integration and interpolation, we have seen that a fixed-step integrator such as the classic fourth-order Runge-Kutta combined with linear interpolation seems the most common choice. This approach completely ignores the question of discontinuities in the velocity field. Some instead use variable-step methods, e.g., Dormand-Prince 5(4) (Romanò and Kuhlmann, 2018) or Runge-Kutta-Fehlberg (Lekien et al., 2003), but even then there is typically no mention of discontinuities, which presumably means that the first strategy listed above has been implicitly chosen. However, this strategy is neither the most accurate nor the most computationally efficient (Hairer et al., 1987, p. 181), so many would most likely benefit from choosing a different approach.

Enright et al. (1988) have developed a procedure following the second strategy. Their approach has the benefit that it requires no a priori knowledge about the discontinuities, but instead uses a procedure that detects discontinuities automatically. This procedure was found to be not only more efficient, but also more accurate than the first strategy listed above. However, the automatic discontinuity detection is not always completely reliable (Enright et al., 1988), so if the locations of the discontinuities are known, the third strategy might be a better choice. The third strategy has also been found to be both faster and more accurate than the first strategy by a considerable amount (Hairer et al., 2008, p. 198). In Nordam and Duran (2020), an approach for stopping and restarting at discontinuities in time was demonstrated. As time, t, is the independent variable, this is fairly straightforward. In this paper, we develop a procedure for handling the discontinuities also in the dependent variable (position). We adapt and build on methods from the mathematical literature, that are perhaps less well-known in the oceanographic community, and we demonstrate quite substantial improvements in numerical accuracy using fairly typical example simulations with modelled ocean currents as input.

#### 3.1 Discontinuities in the temporal dimension

Assuming that the input data is given as snapshots of a vector field at certain known times,  $T_i$ , we know that the interpolated velocity field will have discontinuous derivatives at times  $T_i$ . For a particle trajectory,  $\boldsymbol{x}(t)$ , the time t is the independent variable, and when the discontinuities at  $T_i$  are known it is very easy to stop and restart integration at these times. If the  $T_i$  are equally spaced, that is,  $T_{i+1} - T_i = \Delta T$  for all i, then for fixed-step integrators the simplest strategy is to select a timestep h that divides  $\Delta T$  evenly, and make sure that  $t_0$  coincides with a  $T_i$ , in which case integration will stop and restart at the discontinuities automatically without any further intervention from the integrator. In Nordam and Duran (2020) it was shown

that stopping and restarting integration at the times  $T_i$  gives a substantial improvement in accuracy, and is easily achieved at little cost by simply being mindful of step size selection and the start time.

Suppose that, for some reason, it is not possible to select a step size h that divides  $\Delta T$  evenly, or that the discontinuities  $T_i$  are not evenly spaced. In that case, one must make a small adjustment in steps that are found to cross discontinuities: Assume that at time  $t_n$  the trajectory is at a position  $\boldsymbol{x}_n$ . Then the integrator makes a step of size  $h_n$  from  $\boldsymbol{x}_n$  to  $\boldsymbol{x}_{n+1}$  so that  $t_{n+1} = t_n + h_n$ . If we have  $t_n 

**Figure 2.** Illustration of the boundary crossing procedure with event detection, shown for the fourth-order Runge-Kutta method. The thin dashed lines show the locations of the spatial discontinuities, and the trajectory moves from left to right. When a step is found to cross a discontinuity, Hermite interpolation and bisection is used to find a first approximation of the time of the crossing (top panel). Then a trial step is taken, deliberately a bit short of the boundary, and the Hermite interpolant is used with bisection to extrapolate to find the time of the boundary crossing (middle panel). Finally, a step is taken exactly to the boundary, and a second step to complete the duration of the original timestep (bottom panel). The position of the rightmost point in the bottom panel is more accurate, and thus differs a little from the rightmost point in the top panel.

# Single boundary crossing

Figure 3. Examples of different possible boundary crossings

# 4 Numerical experiments

The purpose of the numerical experiments in this study is to look into how different interpolation schemes affect integration, and to investigate how the integrators with the discontinuity handling described in Section 3.1 perform in comparison to their regular counterparts. We consider a selection of five fixed-step numerical integration schemes from the Runge-Kutta family: One method of order 1 (Euler's method, forward/explicit), one method of order 2 (Heun's method), two methods of order 3 (Heun's method and Kutta's method), and one of order 4 (RK4, or "The" Runge-Kutta method). For details on the methods see Appendix A.

#### 275 4.1 Ocean currents


The current data used in this study was obtained from the Norwegian Meteorological Institute. The datasets were taken from the following model setups:

- Arctic20km (20 km horizontal resolution, 1 h time step),
- Nordic4km (4 km horizontal resolution, 1 h time step),
- NorKyst800m (800 m horizontal resolution, 1h time step).

We have also subsetted the data, to make the file size more convenient, and the data is made available along with the code (Nordam and Mørk, 2025). To allow a direct comparison, we use the same datasets as those used by Nordam and Duran (2020) in the study of special-purpose integrators that stopped and restarted integration at discontinuities in the temporal dimension. The original datasets have dimensions x, y, z and t, though for the purposes of this work only the surface layer has been used. The velocity field is thus interpolated in three dimensions (two spatial dimensions, and time), and we have used the same degree of interpolation in all three dimensions.

The xy plane for all three datasets is defined in polar stereographic projection so that the horizontal plane is a regular (constant spacing) quadratic grid. The current velocity field is given as vector components on the xy basis, and we note that the u and v components have been interpolated to a common grid prior to output (see Fig. 1). We use the xy coordinate system of the polar stereographic projection of the datasets, and we track particle positions in meters in our simulations. This way we can use the vector components directly from the datasets without needing to perform rotation or any other conversion of the data. The error measurements in this study are calculated from Euclidean distances in the xy plane.

## 4.2 Initial conditions






The initial conditions for the trajectory computations are also chosen to be the same as used by Nordam and Duran (2020), namely  $100 \times 100$  points off the coast of Norway, distributed on a regular quadratic grid with grid spacing of 1600 m. The same initial conditions were used for all simulations with all three datasets, and these were chosen to avoid a situation where particles might get stuck in land cells. The particles are generally transported northward, with the eastern half of the particles being subject to more turbulent mixing than the western half. Like Nordam and Duran (2020) we start the trajectories in our simulations at midnight on 8 February 2017, and integrate for 72 h. The initial and final particle positions are shown on a map in Fig. 4, along with the outlines of the subsetted datasets used to force the trajectories.

#### 4.3 Reference solutions

To investigate the performance of the different integration schemes we must estimate the global errors of the computed numerical solutions. Since the true solutions are unknown, we need to estimate the error by purely numerical means, as described in Section 2.1. We compute a set of highly accurate numerical solutions for each of the 10 000 different initial conditions to replace the exact solution x(t) in Eq. (3).

Note that in this study we are not trying to find the solution that most accurately approximates the true trajectories of Lagrangian drifters in the ocean. That is, the exact solution we refer to here is the exact solution with a given realisation of the velocity field, which is not the same solution as we would obtain with true continuous data for the velocity field in the ocean. In Lagrangian oceanography, the aim of interpolation is not to approximate the unresolved turbulent motion of the ocean (which is usually treated by adding random displacements), but rather to have a consistent means for evaluating discrete gridded data at arbitrary positions and times. Different interpolation schemes will provide different values for the current within the grid cells, and hence have a different exact solution to the initial value problem. Therefore, we must compute reference solutions for all combinations of dataset and interpolation scheme. Since we have three different datasets and three different interpolation schemes we thus need nine reference solutions.

For the standard integrators, we computed the reference solutions using the standard fourth-order Runge-Kutta method, and for the integrators with discontinuity handling we used the fourth-order Runge-Kutta method with discontinuity handling. Note also that we computed the optimal step size  $h_{ref}$  for each individual reference solution. For details see Appendix C.

**Figure 4.** The initial positions used in the trajectory simulations, as well as the final positions for the three different datasets. The final positions were obtained with cubic interpolation, the fourth-order Runge-Kutta integrator and a timestep of 60 s. The current speed is shown as a snapshot for the final timestep of the trajectory simulations, and the colour scale is the same for all three subplots.

#### 4.4 Implementation



The code used to run the simulations in this study is available on Zenodo (Nordam and Mørk, 2025), along with the necessary data for the cases shown. It is developed from an earlier code, as described by Nordam and Duran (2020), and we refer to that study for additional details. The code is a simple program for Lagrangian particle transport, written in Fortran. Ocean current data was read using the netCDF library for Fortran, and interpolation was done using the bspline-fortran<sup>2</sup> library. The x and y components of the velocity field were interpolated separately, and the interpolation order was always the same in all three dimensions (x, y, t). We implemented integrators with the procedure for discontinuity handling described in Sects. 3.1 and 3.2, as well as the regular Runge-Kutta integrators.

We note that we use the derived type bspline\_3d from the bspline-fortran library, and for each of the two current components, we create a global interpolator covering the whole simulation in both space and time. This means that when using, e.g., cubic splines, our interpolator is  $C^2$  globally, i.e., it has at least two continuous derivatives everywhere. This is in contrast to for example the local cubic interpolation scheme described by Lekien and Marsden (2005), which is only  $C^1$  globally.

<sup>&</sup>lt;sup>2</sup>https://github.com/jacobwilliams/bspline-fortran (last access: September 1, 2025)

#### 330 5 Results and discussion

We have run trajectory simulations for three different resolution datasets, and three different orders of interpolation, using five different fixed-step integrators from the Runge-Kutta family of methods. In each case, 10 000 trajectories were calculated, from the initial positions shown in Fig. 4. The trajectories were integrated for 72 hours, with timesteps ranging from 120 s to 1 h. We note that all the timesteps are chosen to evenly divide the 1-hour intervals on which the data are provided, such that integration is always stopped and restarted on the discontinuities along the time dimension, as discussed in Section 3.1, and in Nordam and Duran (2020).

#### 5.1 Standard integrators






In Fig. 5, we show the results of running with five regular fixed-step integrators, with orders ranging from 1 to 4 (see Appendix A for details). These results are a baseline, representative of how numerical integration is typically done in Lagrangian oceanography, and are also the same as the results for fixed-step integrators in Nordam and Duran (2020). As mentioned earlier, especially the combination of linear interpolation and fourth-order Runge-Kutta is fairly common, among those studies that provide this information. We note that quintic spline interpolation is probably not used in practice, but it is included here as an example where the order condition in Eq. (4) is satisfied even for fourth-order Runge-Kutta.

The results are shown as work-precision diagrams (see, e.g., Hairer et al. (2008, p. 140)), where the amount of work is indicated by the number of evaluations,  $N_f$ , of the right-hand side of the ODE (average per particle). We note that we use  $N_f$  instead of h as a measure of work partially as it is conventional in the ODE literature, and partially to allow a direct comparison between the standard integrators and the ones with discontinuity handling. For the latter, there is no simple relation between  $N_f$  and h. The precision is indicated by the median relative error, where the relative error is given by

$$E(h) = \frac{|\boldsymbol{x}_N(h) - \boldsymbol{x}_{ref}|}{|\boldsymbol{x}_{ref}|},$$
(8)

0 where  $x_{ref}$  is the highly accurate reference solution (see Appendix C), and the median is taken over all 10 000 particles.

For the standard methods, the number of evaluations of the right-hand side of the ODE is inversely proportional to the timestep, i.e.  $N_f \sim 1/h$ , with a prefactor that depends on the method. Hence, we would expect the error to scale as  $(N_f)^{-p}$ , where p is the order of the method. However, we observe that with linear interpolation, we are only able to achieve second-order convergence, even with higher (third and fourth) order integrators.

The reason we only achieve second-order accuracy in the case of linear interpolation is that when the integrator steps across a spatial boundary, where the first derivative of the velocity field is discontinuous, it picks up a local error that is not bounded by Eq. (4). In Appendix B we show numerically that the local error in stepping across this type of discontinuity is of order  $h^2$ , for all the integrators considered here. Normally, the order of the global error, E(h), is p if the local error is of order p+1: since the number of steps is  $N_t \sim 1/h$ , we have that  $E(h) \sim \mathcal{O}(h^{p+1}) \times N_t = \mathcal{O}(h^p)$ . However, the number of boundary crossings,  $N_b$ , is determined by the length of the trajectory and the resolution of the dataset, and does not depend on the timestep. Hence the effect of the boundary crossings is to add to the global error a term of order  $N_b h^2$ . For the integrators with order p higher

**Figure 5.** Work-precision diagram showing the relative median error as a function of number of evaluations of the right-hand side of the ODE, for five regular fixed-step integrators, ocean currents with three different resolutions, and three different orders of interpolation.

than 2, the error from the boundary crossings will dominate over the global order from the rest of the trajectory, which is of order  $(N_t - N_b)h^{p+1}$ . We see that even for the dataset with 20 km resolution, where only a small fraction of the steps will cross a cell boundary, the lower-order local errors from those steps still dominate the global error (Fig. 5, upper right panel).



With cubic and quintic interpolation, we are able to achieve the expected orders for all interpolation schemes. The reason for this is that the interpolated fields are sufficiently smooth not to introduce any local errors of low enough order to dominate the global error. For cubic interpolation, the third derivative is discontinuous, and as shown in Appendix B, this reduces the local error from order 5 to order 4, for the fourth-order Runge-Kutta integrator. However, since the number of boundary crossings is constant with timestep, the net effect is to add a fourth-order term to the global error, not reducing the order of the overall global error. For quintic spline interpolation, the velocity field has four continuous derivatives, and thus the conditions for Eq. (4) to hold are satisfied for all integrators of order 4 and lower.

**Figure 6.** Work-precision diagram showing the relative median error as a function of number of evaluations of the right-hand side of the ODE, for five fixed-step integrators with discontinuity handling, ocean currents with three different resolutions, and three different orders of interpolation.

#### 5.2 Integrators with discontinuity handling



Figure 6 shows results for the same five integrators, but here with event-detection and handling of the boundary crossings. Here, the integration has been stopped and restarted at each discontinuity at the cell boundaries, as recommended by Hairer et al. (2008, pp. 197–198). We observe that with event detection, we are able to achieve higher than second-order accuracy when using linear interpolation. By stopping and restarting the integration exactly at the boundary we avoid picking up the local errors of order  $h^2$  that stem from the discontinuous first derivative, and thus the global error is of the expected order for each method.

We note that the number of evaluations,  $N_f$ , is now given approximately by  $N_f \sim p T/h + (2p+1)N_b$ , where T is the trajectory duration, p is the order of the method, and  $N_b$  is the number of boundary crossings (which as mentioned depends on the length of the trajectory and the resolution, but not on the timestep). The added constant term in  $N_f$  is the reason that the error as a function of number of evaluations does not form straight lines in the log-log plots in Fig. 6 (this effect is more pronounced for the higher resolution datasets, as these will give more boundary crossings).

**Table 1.** Median relative error for a timestep  $h = 600 \, \text{s}$ , using the fourth-order Runge-Kutta integrator.

| Standard |                        |                        |                        |  |
|----------|------------------------|------------------------|------------------------|--|
|          | 800 m                  | 4 km                   | 20 km                  |  |
| Linear   | $3.69 \times 10^{-8}$  | $2.92 \times 10^{-9}$  | $6.88 \times 10^{-10}$ |  |
| Cubic    | $3.33 \times 10^{-10}$ | $6.00 \times 10^{-12}$ | $2.21 \times 10^{-12}$ |  |
| Quintic  | $1.29 \times 10^{-10}$ | $2.06 \times 10^{-11}$ | $2.39 \times 10^{-11}$ |  |





| Event detection |                        |                        |                        |
|-----------------|------------------------|------------------------|------------------------|
|                 | 800 m                  | 4 km                   | 20 km                  |
| Linear          | $8.6 \times 10^{-10}$  | $3.58 \times 10^{-12}$ | $6.34 \times 10^{-13}$ |
| Cubic           | $2.42 \times 10^{-10}$ | $7.99 \times 10^{-12}$ | $2.36 \times 10^{-12}$ |
| Quintic         | $2.15 \times 10^{-10}$ | $5.72 \times 10^{-11}$ | $3.25 \times 10^{-11}$ |

**Table 2.** Runtime in seconds needed to achieve a median relative error of  $10^{-10}$ , using the fourth-order Runge-Kutta integrator. <sup>†</sup>For the 800 m resolution dataset with linear interpolation, this accuracy was not achieved with the standard method, hence this runtime is based on linear extrapolation of the error as a function of runtime in a log-log plot (see Fig. D1.)

| Standard |                            |                   |        |
|----------|----------------------------|-------------------|--------|
|          | 800 m                      | 4 km              | 20 km  |
| Linear   | $150.2~\mathrm{s}^\dagger$ | 45.8 s            | 22.1 s |
| Cubic    | 42.7  s                    | $15.5 \mathrm{s}$ | 11.9 s |
| Quintic  | 94.0 s                     | 59.0 s            | 61.2 s |

| Event detection |                    |                   |         |
|-----------------|--------------------|-------------------|---------|
|                 | 800 m              | 4 km              | 20 km   |
| Linear          | 26.2 s             | 6.4 s             | 3.1 s   |
| Cubic           | 68.1 s             | $24.7 \mathrm{s}$ | 14.3  s |
| Quintic         | $175.1 \mathrm{s}$ | 94.1 s            | 71.6 s  |

For cubic and quintic interpolation, the accuracy achieved with the event-detection method is essentially identical to that obtained with the standard methods. As discussed in Section 5.1, this is because the cubic and quintic interpolants are sufficiently smooth that the standard integrators operate as expected.

Comparing the standard integrators and the event-detection method, we see that the main difference is found when using linear interpolation, and an integrator of order 3 or 4. The combination of linear interpolation and fourth-order Runge-Kutta appears in many studies in applied oceanography, making this a very relevant case to discuss. In Table 1 we show the median relative error obtained with a 600 s timestep with the fourth-order Runge-Kutta integrator, for all nine combinations of resolution and interpolation orders, after 72 hours of integration. For linear interpolation, the error with the 600 s timestep is 1.5–3 orders of magnitude smaller with the event detection approach. With a shorter timestep, the difference becomes even greater due to the higher order achieved with event detection, but in practice timesteps much shorter than 600 s are probably rarely used in applied oceanography. With cubic or quintic interpolation, the errors are very similar between the standard and the event detection approaches, or even a bit worse with event detection (presumably due to the larger number of floating point operations).

Another way to consider the results is to look at the runtime needed to achieve a given level of accuracy. In Table 2, we show the runtime needed to achieve a median relative error of  $10^{-10}$  after 72 hours of integration, which is about the accuracy obtained with fourth-order Runge-Kutta, cubic interpolation, and a timestep of h = 600 s. To find the runtime needed for a given error, linear interpolation was applied to error as a function of runtime on a log-log scale (see Fig. D1 and D2). We see that for cubic and quintic interpolation, the method with event detection is slower, due to the additional number of operations involved in crossing cell boundaries. For linear interpolation, however, the event detection method achieves the same accuracy as the standard approach, but with a factor 6–7 reduction in runtime. For higher accuracy (smaller error), the difference will be even larger.

# 405 5.3 Application to backtracking





A common application of Lagrangian particle methods is in backtracking, used for example to find the source of an oil spill (Galt and Payton, 1983; Suneel et al., 2016), the origins of plastics in the ocean (Strand et al., 2021; van Duinen et al., 2022), or the site of an accident in search and rescue (Abascal et al., 2012; Drévillon et al., 2013; Breivik et al., 2025). A somewhat subtle element in backtracking is that simulations often start from observations, where floating objects have been found. Floating objects at sea are not uniformly distributed, but are more likely to be found in zones of convergent surface currents (D'Asaro et al., 2018). In backtracking, one essentially reverses the velocity field, such that zones of convergence become zones of divergence. This provides two challenges: First, when a particle is found in a convergence zone, it is typically not known when it arrived in that zone. When reversing the velocity field in the backwards simulation, the particle may leave the convergence zone at a different time than when it arrived. Second, when a particle is in a convergence zone, even a small error in the position can lead to a large difference in the direction of the velocity field. Both of these effects can lead to large errors in backtracking, as discussed in Reijnders et al. (2024) and Breivik et al. (2025). Any initial errors will tend to grow over the course of a trajectory, and for these reasons it may be of extra importance to use accurate numerical schemes in backtracking.

Just to be clear, we note that there is no intrinsic difference between forwards and backwards trajectory modelling. Aside from some asymmetries such as river outlets, the reversed velocity field of the ocean is not drastically different from the original field. The challenge in backtracking is thus not due to the velocity field, but rather due to the initial positions, which tend to be located in zones of convergence (D'Asaro et al., 2018).

To test our numerical schemes on a realistic case of backtracking, we first start out 2500 particles in a  $50 \times 50$  grid of initial positions, and run these forwards in time for 25 days, using the 4 km resolution currents, linear interpolation, and the fourth-order Runge-Kutta method both with and without discontinuity handling, with a 10 s timestep. We note that we consider transport in the horizontal plane, using only the surface currents. The initial and final particle positions are shown in Fig. 7, and we observe that many of the particles are found in relatively narrow bands of high concentrations, corresponding to the convergence zones discussed above.

Using the final positions of the forward runs as initial positions, we conduct backtracking with the same dataset and interpolation (4 km, linear). We use the fourth-order Runge-Kutta method, with and without the discontinuity handling scheme described in Section 3.2, and we use a range of timesteps from 60 s to 3600 s. In Fig. 8, we illustrate the results qualitatively. If the backtracking was perfect, all particles would end up back inside the initial square (dashed black lines). We see that shorter timesteps are better, and we also see that the method with discontinuity handling does a better job than the standard method.

Finally, in Fig. 9, we show the median and mean error as a function of timestep. Here, the error is defined as the absolute distance away from the true initial location we are trying to recover by backtracking. As we expect from the discussion in Sections 5.1 and 5.2, we see second-order convergence for the median error of the standard fourth-order Runge-Kutta method, due to the discontinuous derivatives of the linear interpolation. The method with event detection and discontinuity handling

**Figure 7.** Particles from the initial position (grey square) are modelled forwards in time for 25 days (black dots), using modelled surface currents with 4 km horizontal resolution, linear interpolation, and the fourth-order Runge-Kutta method with discontinuity handling. The final positions of the forwards modelling are then used as initial conditions for an exercise in backtracking. The dashed line shows the outline of the subsetted data, which is provided along with the code (Nordam and Mørk, 2025).

is able to achieve fourth-order convergence, and the median error at a timestep of 600 s is around three orders of magnitude smaller than for the standard method.

For the mean error, the convergence is much slower, which is more unexpected. We see from the illustration in Fig. 8 that most particles end up very close to the correct position, while a small percentage (up to 20% for the longest timestep and the standard method) end up quite far away from the correct positions, outside the dashed square. The median error is dominated by the majority of particles with a fairly small error, while the few particles with an error of 100 km or more will dominate the average. We conclude that the growth in the average error is driven primarily by the growth in number of particles that have very large errors. The responsible mechanisms are presumably the timing of the departure from the convergence zones in the backwards run, as well as the initial direction on leaving the convergence zones, as discussed above. While both methods show slower convergence of the mean error, the method with discontinuity handling still performs far better than the standard method, with faster convergence and about 1 order of magnitude lower error for a timestep of 600 s, and the improvement is even better for shorter timesteps.

**Figure 8.** Result of the backtracking, for the fourth-order Runge-Kutta method with and without discontinuity handling, and for three selected timesteps. Also indicated is the number of particles that are more than 0.1 m outside the initial square. If the backtracking was perfect, all particles would end up inside the black dashed line.

**Figure 9.** Median error (top panel) and mean error (bottom panel), relative to the ideal initial position, as a function of timestep, for the fourth-order Runge-Kutta method with and without discontinuity handling.

#### 450 5.4 Comparison to variable-timestep methods



The topic of Nordam and Duran (2020) was discontinuities in the partial derivatives of the right-hand side of the ODE along the time dimension. Two strategies were investigated: First using fixed-step integrators with timestep and start time chosen such that the integration always stops and restarts exactly at the times when the input data is defined, and second to use variable-timestep integrators, but modify them in such a way that they stop exactly at these times. The first strategy from Nordam and Duran (2020) is the same as what we have presented in Section 5.1. Here, we present a comparison of the variable-step methods from Nordam and Duran (2020) to the fixed-step integrators with event detection and spatial discontinuity handling developed in the current study. We limit the comparison to linear interpolation, as this is both the most common use case, and the case where the greatest differences are seen.

We have not discussed variable-step methods so far, but briefly they are ODE methods where one specifies an error tolerance instead of a timestep, and then the method will take as short steps as needed to meet the tolerance. Variable step methods are usually very efficient for general ODE problems (Hairer et al., 2008), but do not appear to be widely used in Lagrangian

**Figure 10.** Median relative error after 72 hours trajectory duration, for the three variable-timestep methods discussed in Nordam and Duran (2020), and for the fourth-order Runge-Kutta method with event detection and spatial discontinuity handling. For the variable-timestep methods, the continuous lines show the standard methods, and dashed lines show the special methods with temporal discontinuity handling.

oceanography. The methods we consider are Bogacki-Shampine 3(2) (Bogacki and Shampine, 1989), Dormand-Prince 5(4) (Dormand and Prince, 1980), and Dormand-Prince 8(7) (Dormand and Prince, 1986). In Fig. 10, we show both the standard variants, and the special variants developed in Nordam and Duran (2020), which stop and restart the integration at the discontinuities in time. For a detailed description, see Nordam and Duran (2020).




In Fig. 10, we show the median relative error after 72 hours of transport (as shown in Fig. 4), for the fourth-order Runge-Kutta method with event detection and spatial discontinuity handling, compared to three variable-step methods. As the variable-step methods do not use a fixed timestep, we again compare the integrators by plotting error as a function of number of evaluations of the right-hand side,  $N_f$ , as a measure of the work. Broadly speaking, the simulation runtime is proportional to  $N_f$ , and for a fixed-step method such as fourth-order Runge-Kutta,  $N_f \sim 1/h$ , where h is the timestep. As described in Section 3.2, for the method with event detection and discontinuity handling, some additional evaluations are done each time a cell boundary is crossed.

The timesteps considered for the fourth-order Runge-Kutta method range from 90 to 3600 s, and we see that the method with event-detection and (spatial) discontinuity handling compares favourably to the varible-step methods, both the standard ones and the ones that handle discontinuities along the time dimension. For the dataset with 800 m resolution, corresponding to the largest number of boundary crossings during a 72 h trajectory, the event-detection method gives better efficiency (smaller error for the same amount of work) for timesteps of 1800 s or shorter. If lower accuracy is acceptable, the Bogacki-Shampine 3(2) or Dormand-Prince 5(4) methods, with temporal discontinuity handling may be preferable.

For the dataset with 4 km horizontal resolution, the Runge-Kutta method with event-detection and spatial discontinuity handling gives better efficiency than the variable step methods for all timesteps considered. For the 20 km dataset, Dormand-

Prince 5(4) or 8(7) may give better efficiency for very high accuracies, corresponding to timesteps shorter than 600 s for the Runge-Kutta method.

Overall, we find the fixed-step fourth-order Runge-Kutta method with spatial discontinuity handling (and a timestep chosen to handle temporal discontinuities) to be preferable to variable-step methods with temporal discontinuity handling only. Even though the latter may strictly give better efficiency in some cases, the differences are small, and may be outweighed by the convenience of having a fixed timestep. With regards to complexity of implementation, the two methods are comparable.

## 5.5 Comparison to other methods







We note that our method for discontinuity handling with event detection has some similarities to the analytical method employed by models such as ARIANE (Blanke and Raynaud, 1997) and TRACMASS (Döös et al., 2017; de Vries and Döös, 2001). These models do not operate on a timestep in the traditional sense, using instead an event-driven approach. When a particle enters a "cell" defined by the grid of the hydrodynamic data, an analytical equation is solved to find the time and location where the particle will leave that cell, assuming linearly interpolated currents. Effectively, this stops and restarts the integration exactly at the cell boundaries, thus avoiding the lower-order error terms. With the event-detection method we presented in Section 3.2, the integration is also stopped and restarted at the boundary, except that the time at which the particle reaches the boundary is determined numerically instead of analytically. In particular, if the integration timestep is chosen identical to the timestep the data is provided on,  $h = \Delta T$ , then the event detection method will stop and restart integration at all cell boundaries in both space and time, and nowhere else, which has strong similarities with the analytical method.

The analytical method employed by, e.g., TRACMASS and ARIANE, has been found to yield very low numerical errors. This makes it particularly suited for global-scale water mass transport studies as evidenced, for example, by only small differences in diagnosed inter-ocean mass transfers obtained from forward and backward calculations (Blanke et al., 2001). Such practical reversibility of forward and backward calculations is not necessarily obtained by standard time-stepping methods, as seen for example in Reijnders et al. (2024). In that study, particles are tracked backwards, and then forwards, and found to return to their original position with far higher accuracy when using the analytical method, compared to standard fourth-order Runge-Kutta and linear interpolation. Similarly, we show in Section 5.3 that our modified Runge-Kutta method with spatial discontinuity handling performs far better than the standard method, when tested on a case of backtracking with linearly interpolated data.

While we have not directly compared the accuracy of our event-detection approach to the analytical method, we do believe that our event-detection method offers the advantages of relatively simple implementation and great flexibility regarding the velocity input velocity data as well as the choice of interpolation and integration schemes. Our method is grid-agnostic, and can use data that are natively on either A-grid or C-grid, as well as data that has been produced on a C-grid, and then interpolated to cell centers (which is very common for openly available velocity products such as those from Copernicus Marine Service). Our method is also adaptable to unstructured grids, the only requirements are that we can tell when a trajectory crosses a cell boundary, and that we need a function that has a root at the cell boundary (see Eq. (5)), which can be used with the bisection scheme to find the time of the boundary crossing. Our method is also interpolation agnostic, and can be used for

example with the local cubic spline interpolation of Lekien and Marsden (2005). In short, our method provides the opportunity to combine some of the previously distinct features of analytical and standard time-stepping methods for the calculation of particle trajectories in the marine environment.

# 6 Conclusions





This study builds on a previous study by Nordam and Duran (2020), which demonstrates that stopping and restarting integration at discontinuities in time can improve accuracy. Here, we have extended the idea by stopping and restarting integration at discontinuities in the spatial dimensions as well as in time. To identify the times at which spatial cell boundaries were crossed required the use and adaptation of methods from the general numerical literature, and we developed a variant of a procedure described by (Hairer et al., 2008, pp. 195–198) which was found to work well.

We have tested the developed procedure on some typical applications in Lagrangian oceanography, and we demonstrate that with our method for discontinuity handling we achieve fourth-order accuracy with fourth-order Runge-Kutta and linarly interpolated currents. The standard fourth-order Runge-Kutta is only able to achieve second order accuracy with linearly interpolated currents, due to the discontinuous derivatives at every cell boundary. When combined with higher-order interpolation schemes, the results are essentially the same regardless of integration method, though in practice linear interpolation is the most commonly used approach in oceanography, due to its volume conserving properties.

As a comment on the standard methods, we note that the combination of fourth-order Runge-Kutta and linear interpolation is not an optimal choice, despite being the most common. As shown in Fig. 5, the second-, third-, and fourth-order standard methods all show second-order convergence only, and in fact the third-order method due to Kutta has a slightly smaller error for the same effort across a large range of timesteps, compared to the standard fourth-order method.

Studies where additional accuracy may be important include backtracking for source attribution (Strand et al., 2021) or search and rescue (Breivik et al., 2025), and generally any study with long integration times. Using integrators with discontinuity handling can considerably reduce the numerical error in such simulations. Alternatively, the same accuracy as with standard methods can be achieved with much reduced computational effort. The complexity of the implementation is modest, and the fact that the modified integrators with discontinuity handling still use a fixed timestep has some practical advantages when it comes to reading of input data, and writing of output.

Code and data availability. The code used to run the simulations and analyse the results, as well as the ocean current data used in this study, is published at Zenodo (https://doi.org/10.5281/zenodo.15781959; Nordam and Mørk, 2025), where there is also a link to the GitHub repository. The code is published under a GNU GPL-3.0 license.

The simulation code is written in Fortran and built using CMake. It requires three external libraries: netCDF, hdf5, and bspline-fortran. Analysis of simulation results is done in Jupyter Notebooks.

# 545 Appendix A: Runge-Kutta methods

In this study we consider a selection of five fixed-step numerical integration schemes from the Runge-Kutta family. We consider one method of order 1 (Euler's method), one method of order 2 (Heun's method), two methods of order 3 (Heun's method and Kutta's method), and one of order 4 (the classic fourth-order Runge-Kutta).

With a general s-stage explicit Runge-Kutta method, the next step from position  $x_n$  at time  $t_n$  to position  $x_{n+1}$  can be expressed as (see, e.g. Hairer et al., 2008, p. 134)

$$x_{n+1} = x_n + h \sum_{i=1}^{s} b_i k_i, \tag{A1}$$

where the coefficients  $k_i$  are given by

$$k_i = f\left(x_n + h\sum_{j=1}^i a_{ij}k_j, t_n + hc_i\right). \tag{A2}$$

The coefficients  $a_{ij}$ ,  $b_i$ , and  $c_i$  for a specific method can be expressed in a so-called Butcher tableau, after Butcher (1964).

The Butcher tableau for the general method represented by Eqs. (A1) and (A2) is given in Table A1. We also present the Butcher tableaus for the specific methods used in this paper in Table A2.

**Table A1.** Butcher tableau for a general s-stage Runge-Kutta method represented by Eqs. (A1) and (A2).

# Appendix B: Error in boundary-crossing steps

When stepping across a spatial boundary between cells, during integration of a Lagrangian trajectory, we pass a point where f(x,t), or some of its derivatives, are discontinuous in x. In the case of linear interpolation, the function itself is continuous, but the first derivative is a discontinuous step function. To illustrate the error in stepping across such a point, we consider the continuous, piecewise linear function

$$f(x) = \begin{cases} 1+x & \text{if } x 

**Figure B1.** Absolute value of the local error, for a single step of length h. In the left panel stepping across a point with discontinuous first derivative (see Eq. (B1)), and in the right panel without the discontinuity. We note that all integrators have a local error of order  $h^2$  when stepping across the discontinuity, and their expected orders when there is no discontinuity. For this particular example, the local error of the classic fourth-order Runge-Kutta and Kutta's third-order method are almost identical, which is why the green line is not visible.

In Fig. B2 we show the result of a similar test as described above, but where the right-hand side of the ODE is given by the continuous, piecewise cubic function

$$f(x) = \begin{cases} x + (x-1)^3 & \text{if } x 

Figure B2. Absolute value of the local error, for a single step of length h. In the left panel stepping across a point with discontinuous third derivative (see Eq. (B3)), and in the right panel without the discontinuity. We note that no integrator has a local error of order higher than  $h^4$  when stepping across the discontinuity, while for the case with no discontinuity the fourth-order method has the expected fifth-order local error.

as we make an increasing number of steps, and therefore an increasing number of calculations (Lapidus and Seinfeld, 1971, pp. 35–37). Clearly, there exists an optimal timestep, where the sum of the roundoff error and the truncation error is minimal.



To obtain the numerical reference solutions for this study we wish to use a timestep close to the optimal. To estimate the optimal timestep, still in the absence of analytical solutions, we use an approach of repeatedly halving h, with a measure of the error given by

$$\tilde{E}(h) = |\boldsymbol{x}_N(h) - \boldsymbol{x}_{N/2}(2h)|. \tag{C1}$$

Here,  $\tilde{E}(h)$  is the difference between the solution after N steps with timestep h, and the solution after N/2 steps with timestep 2h. We average over all particles, and plot the measure of the error as a function of timestep, as shown in Fig. C1. The timestep where the difference between the error at h and at h is smallest, will be used as the reference.

The timesteps used for the reference solutions are summarised in Table C1. The different datasets, and also the different interpolation schemes, in general give different velocity vector fields. Hence, they also give different solutions, and separate reference solutions are obtained for each of the nine combinations of dataset (resolution) and interpolator. The reference solution is calculated with the fourth-order Runge-Kutta integrator in all cases, using standard RK4 for the standard methods, and RK4 with discontinuity handling as reference for the methods with discontinuity handling.

**Figure C1.** A measure of the error,  $\tilde{E}(h)$  (see Eq. (C1)), as a function of timestep, for the standard integrators (left) and the integrators with event detection (right). All results are for fourth-order Runge-Kutta.

**Table C1.** Timesteps used to obtain the numerical reference solutions for the error analysis, for the standard integrators (left) and the integrators with event detection and discontinuity handling (right). In all cases, the fourth-order Runge-Kutta integrator is used for the reference solution.

| Standard |       |      |       | Discontinuity handling |       |      |       |
|----------|-------|------|-------|------------------------|-------|------|-------|
|          | 800 m | 4 km | 20 km |                        | 800 m | 4 km | 20 km |
| Linear   | 10 s  | 10 s | 10 s  | Linear                 | 10 s  | 20 s | 60 s  |
| Cubic    | 10 s  | 30 s | 60 s  | Cubic                  | 10 s  | 30 s | 60 s  |
| Quintic  | 10 s  | 30 s | 30 s  | Quintic                | 10 s  | 30 s | 30 s  |

**Figure D1.** Median relative error as a function of runtime, for the standard integrators. The dotted, dash-dotted and dashed lines show expected behaviour for respectively second, third and fourth-order methods, if runtime was inversely proportional to the timestep.

# Appendix D: Additional results

In Figures D1 and D2 we show work-precision diagrams where the error is shown as a function of runtime,  $T_{\rm run}$ . The runtime was measured on an Intel i9 CPU (3.7 GHz), with the program running on a single core. Reading of current data from disk, and creation of the interpolator object, was not included in the measured time.

**Figure D2.** Median relative error as a function of runtime, for integrators with event detection. The dotted, dash-dotted and dashed lines show expected behaviour for respectively second, third and fourth-order methods, if runtime was inversely proportional to the timestep.

Author contributions. JMM developed the method. The method was implemented by JMM and TN into simulation code previously written by TN. Simulations were run by TN. SR contributed to the discussion of the results. JMM and TN wrote the manuscript with additions and editing by SR.

Competing interests. The authors declare that they have no conflict of interest

Acknowledgements. The authors would like to thank Daan Reijnders, Erik van Sebille, Øvind Breivik, Knut-Frode Dagestad and Raymond Nepstad for good discussions.

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
