# Peer review of "Handling discontinuities in numerical ODE methods for Lagrangian oceanography"

_EGUsphere, 2025_

## Author Response (AR1)

**Response to reviewers**

Dear Editor, dear Reviewers,

Thank you for your comments and questions concerning our manuscript. We have carefully reviewed each comment and addressed them with appropriate changes to the manuscript. We have also made a few additional changes to generally improve the manuscript.

Throughout this document, you will find the reviewer comments as indented blocks in italics, and our response in an upright font below.

Sincerely,

J.M. Mørk, T. Nordam, and S. Rühs

**Anonymous Referee #1**

The manuscript by Mork et al. offers a method to handle discontinuities when interpolating the velocity fields in Lagrangian modeling. I find this method interesting and useful, and it should be used in many Lagrangian models.

This manuscript also includes a survey of how different interpolation methods and timestep (fixed vs. variable) were used in previous studies. The authors also showed how different methods of discontinuity handling interact with interpolation methods to affect simulation accuracy. The authors tested how their method can improve the accuracy in a backtracking setting.

While I am not an expert on this topic, I cannot find any problems with the maths presented in this paper.

We thank the referee for the positive comments on our manuscript.

While I find the science itself is robust, I believe the presentation structure can be improved. I recommend that the authors to have a methodology section that can combine the sections of theory and numerical experiments, highlighting the authors' real contributions.

We thank the reviewer for the suggestion, however we feel that the current structure with separate sections for Theory, Implementation, and Numerical experiments, is clearer than if we combine all these into a methodology section.

Regarding our contribution, we feel that taking methods from the mathematical literature and making them more accessible to an oceanographic audience, with the use of relevant demonstration cases, is an important point. We have attempted to further highlight this, and our other contributions, by describing in more detail how we made adaptations to the methods we used, in Sections 3, 3.2, and 6.

We also made a small restructuring of the results/discussion and the conclusion, as described below.

Also, I find the conclusion section too long with too much discussion and additional results. These extra results and discussion should be moved to the Results and Discussion section. The conclusion section itself should be concise, highlighting the main take-home messages.

We agree that Tables 1 and 2 more properly belong in the results section, as they are an alternative way of presenting some of the results from Figs. 5 and 6. We have moved the tables and the discussion thereof into Section 5.

We have also attempted to streamline the conclusion a bit, to highlight what is new, and why it is important.

A final minor comment is that the "??" should be resolved in line 375.

This has been fixed.

**Willi Rath, Referee #2**

In the manuscript "Handling discontinuities in numerical ODE methods for Lagrangian oceanography", JM Mørk et al. address the problem that in Lagrangian simulations based on spatially interpolated velocity fields with interpolation schemes of limited smoothness, the order of accuracy of the solution is not determined by the time stepping scheme but also by the order of differentiability of the interpolation scheme. They do so by developing, implementing and testing time stepping schemes which adapt spatial step sizes in such a way that the locations at which discontinuities in the interpolation scheme are exactly met, hence avoiding the approximation of a non-smooth velocity field with a smooth function.

The manuscript addresses an important topic in numerical oceanography and provides a possible solution in a way that is ready to be picked up by the community. So I recommend publishing the manuscript after minor revisions most of which are related to the presentation of the results.

A note on reproducibility: The code and data provided by the authors made it very easy to check aspects of the results myself. I especially appreciate the fact that the manuscript only relies on small but well chosen velocity datasets.

We thank Dr. Rath for the positive comments, as well as for taking the time to look into the code and the datasets used. We made an effort to make our results reproducible, and we are happy to get confirmation that this is indeed possible.

Minor remarks:

The scaling figures use three different x-axes (e.g. Fig. 5 uses log of number of evaluations, Fig 9 uses time step but on a nonlinear (log?) scaling, Fig. B1 uses log of time step). While reading, I found myself translating the time step into num of evals and back. So please consider making this easier for the reader. (Maybe by providing both a time and num-of-evals axis?)

We realise that this might seem a bit confusing, but we believe there are good reasons to keep this the way it is.

For Figs. 5 and 6, we use number of evaluations partially because this is conventional (see, e.g., Hairer, Nørsett & Wanner), and partially to facilitate a direct comparison between the standard methods and the methods with discontinuity handling. In the latter case, there is no simple relationship between timestep and the number of evaluations, as there are extra evaluations at each boundary crossing (how many depend on the order of the method). Hence, it is not possible to include both values, for the methods of different order, on the same axis. While the results could have been plotted as a function of timestep, we prefer to keep it as a function of number of evaluations. We have attempted to make this more clear in the text.

In Fig. 9, the results are presented as a function of timestep, on a logarithmic scale. It is standard logarithmic, but for pedagogical reasons we chose to use "round values" of the timestep in seconds as the ticks, instead of 100, 1000, etc. Clearly, Fig. 9 could also have used number of evaluations as a measure of work, but in this case we feel timestep is better suited. As this is backtracking, the measure of the error is absolute (distance away from the actual initial location we are trying

to recover), rather than relative, as for Figs. 5 and 6, and we feel that it in this case is easier to relate to the timesteps and the absolute error.

Finally, for Figs. B1 and B2, these show the local error, i.e., the error in a single step. Hence, these cannot be plotted as a function of number evaluations, as this is a constant for each method.

Some of the scaling figures (e.g. Fig. 5, 9, B1) provide reference lines for powers of the time step and some don't. I suggest adding the reference lines where they are currently absent.

Reference lines were already present in Figs. 5, 6, 9, B1 and B2, but we have now added a line also in Fig. 10. Note that the legend in Figs. 5 and 6 are split across two subplots, to avoid overlapping the plotted lines.

Fig. 4 could benefit from some transparency (alpha ; 1 - possibly also need to adapt the edge alpha) for the final positions. This would improve visibility of the convergent regions.

We have replotted Fig. 4, using a dashed outline instead of individual markers to show the initial position, using a smaller alpha for the final positions, and switching to dot markers with no outline. We also chose to add a snapshot of the current speeds here, as this helps highlight the differences seen in the transport between the datasets.

I found myself downloading the data provided by the authors to explore statistics of the velocity fields used in the study: I checked max. velocities to add context to the time steps used in the study. I also checked alignment of of velocities between neighbouring grid boxes and speed differences between neighbouring grid boxes. While it's not necessary to include all of this in the paper, a plot showing a snapshot of speed or velocity for the three datasets to provide info on the dynamic regime for context could help. When doing Lagrangian simulations based on velocities of finite volume grids, we often think of the velocities as fluxes only defined on the faces of the finite volume grid cells. To accurately represent how a particle would move in this velocity field, we're typically limited to linear interpolation (as e.g. described in Delandmeter and van Sebille, 2019, https://doi.org/10.5194/gmd-12-3571-2019). So the combination of low-order interpolator, high-order time stepping with discontinuity handling is the only approach left.

Many thanks for taking the time and making the effort to looking into our data. As mentioned above, we have included snapshots of the current speed in Fig. 4.

We also appreciate the point about linear interpolation having some fundamental advantages over higher-order methods. We have made a mention of this in section 3 in the revised manuscript:

"A solution could thus be to combine a high-order ODE method with an interpolation scheme of sufficiently high order. However, higher-order spline interpolation is more computationally demanding without necessarily being a more faithful representation of the underlying ocean dynamics. In fact, linear interpolation may be required for particle trajectory simulations based on (i) finite-volume model output or (ii) finite-difference model output, when trajectories are intended to represent volume transport pathways. In these cases, the velocity reconstruction between cell faces must preserve the discrete flux balance, which implies a constant divergence within each grid cell. Linear interpolation satisfies this condition, whereas standard higher-order schemes (e.g., cubic or spline interpolation) generally do not. This suggests that a better approach might be to use low-order interpolation together with higher-order ODE methods, but somehow try to handle the discontinuities."

Thoughts on the appendices (please take as suggestions and not hard requirements):

I am not sure if the whole appendix A on time stepping schemes is really necessary. While the explicit presentation Butcher tables used in the manuscripts helps avoiding

confusion about the different Heun methods, 1 1/2 pages of text and discussions and tables for just presenting the coefficients feels excessive.

We prefer to keep the Butcher tableaus to be explicit about the methods we have used, but we appreciate the point about the space. We have made the presentation more compact by presenting all tableaus in the same table, which we believe will look reasonable in the final two-column layout.

I am also not sure if Appendix D is really necessary. As you basically eliminated the data loading and interpolator setup from the timings, these results look very similar to Fig. 5. But if the paper is read also in the computational physics / computer science community, scaling of runtimes might be their first concern.

We would prefer to keep Appendix D as well, partially to allow a direct comparison with the earlier paper this study builds on (Nordam & Duran, 2020).